# New Standard for Metal Powder Bed Fusion Surface Texture Measurement and Characterisation

**Adam Thompson** [1,*] , **Lewis Newton** [1,2] **and Richard Leach** [1]

1 Manufacturing Metrology Team, Faculty of Engineering, University of Nottingham, Nottingham NG7 2RD, UK; lewis.newton2@the-mtc.org (L.N.); richard.leach@nottingham.ac.uk (R.L.)
2 The Manufacturing Technology Centre Ltd., Pilot Way, Ansty Park, Coventry CV7 9LU, UK
* Correspondence: adam.thompson@nottingham.ac.uk

**Abstract:** As metal additive manufacturing has been increasingly accepted as a viable method of industrial manufacture, there has been a significant uptake in manufacturers wishing to verify and test their parts through analysis of part surface. However, various studies have shown that metal additive surfaces tend to exhibit highly complex features and, thus, represent a challenge to those wishing to undertake measurement and characterisation. Over the past decade, good practice in metal additive surface measurement and characterisation have been developed, ultimately resulting in the creation of a new standard guide, ASTM F3624-23, which summarises that good practice. Here, we explain the background and rationale for the creation of this standard and provide an overview of the contents of the standard. An example case study is then presented, showing the worked good practice guidance in a metal additive surface measurement and characterisation task, namely, a comparative measurement of an example surface using two different instruments. Finally, considerations for future versions of the standard are presented, explaining the need to develop further good practice for novel instruments and to focus on feature-based characterisation approaches.

**Keywords:** metrology; surface texture; additive manufacturing; laser powder bed fusion; measurement; characterisation





## 1. Introduction

In the past decade, significant advances have been made in the measurement and characterisation of metal additively manufactured (AM) surface topography, with papers having been published on various aspects of the topic by a wealth of authors and research teams [1–30]. The first major review on this topic was conducted by Townsend et al. in 2016 [7], in which the authors highlighted a number of significant challenges in metal AM surface measurement and characterisation. Particularly, Townsend et al. noted that the majority of characterisation activities at that time were being performed on profile measurements, with the ISO 21920-2 [31] *Ra* parameter being most commonly used to perform characterisation. Townsend et al. further noted, as did Senin et al. [2], that areal measurement and characterisation methodologies presented an opportunity to improve measurement and characterisation of metal AM surfaces. These authors also noted that feature-based characterisation methodologies offered significant potential improvement to the value of performing metal AM surface measurement, with feature-based approaches providing new information regarding process phenomena that could not be captured by field parameters [2,3,7].

Townsend et al. [7] concluded that, within the realms of metal AM surface measurement and characterisation, results were often not fully reproducible, as key information was not reported. Additionally, they found that while some research teams were beginning to put effort into understanding the complex features present on metal AM surfaces, significant work was required in the development of instrument optimisation and characterisation pipelines, as well as in establishing correlations between part function and surface texture.

Following the publication of the Townsend et al. [7] review, significant effort was undertaken by the research community to address the issues they found, particularly in the development of instrument optimisation and characterisation pipelines. This work began with a number of studies examining the features present on metal AM surfaces. For example, Thompson et al. [3] explored the various features present on metal surfaces, noting the presence of "relevant topographic detail at multiple scales, with a mixture of high and low aspect-ratio formations, high slopes, undercuts and deep recesses". Most of the work performed thus far has been concerned with the measurement of metal powder bed fusion (PBF) surfaces [7]. The two types of PBF, laser beam (PBF-LB) and electron beam (PBF-EB), are amongst the most extensively adopted metal AM technologies, but also provide the most complex surfaces [7]. Despite this complexity, during the past decade, the questions of "what features exist on PBF surfaces" and "how do we measure PBF surfaces" have largely been answered in the various publications discussed here.

However, while the research community has made notable strides forward in understanding metal AM surfaces, translation of the information gained in the research landscape onto the industrial shop floor is an ongoing task. There have been extensive efforts in the standardisation of both surface texture measurement and characterisation (via the ISO 21920 [32] and 25178 [33] series of standards) and AM (via the ISO 529XX [34] series of standards) but there has, until recently, not been standardisation in good practice for surface measurement for AM. The need for standardisation in this area has become clear in the last five years, with ASTM raising a work item [35] in 2019 as a result of increasing industrial pull for such a document.

Following extensive work in the research community and the activities of ASTM, we recently finalised the publication of an appropriate standard, *ASTM F3624-23 Standard Guide For Additive Manufacturing Of Metals—Powder Bed Fusion—Measurement And Characterization Of Surface Texture* [36]. In this brief report, we provide a short summary of the good practice guidance presented in ASTM F3624-23 alongside an example of this good practice applied in a case study.

## 2. Good Practice in Measurement and Characterisation of Metal PBF Surfaces

ASTM F3624-23 is designed to provide an introduction to surface texture measurement and characterisation of surfaces manufactured using metal PBF, providing reference to existing standards, where appropriate, and explicit guidance where none is otherwise available.

The standard is explicitly aimed at industrial users of measurement instruments and characterisation software, who may or may not have experience of surface texture metrology. The standard begins with an overview of general concepts, covering surface texture metrology within the context of the standardisation landscape and wider industry. The standard then contains a summary of different methods of surface measurement, particularly profile and areal paradigms (via the ISO 21920 [32] and 25178 [33] series of standards, respectively). The standard includes an explanation of filtering methodologies (in both profile and areal cases) and provides general considerations that must be made during metal PBF surface measurement, particularly relating to the following measurement challenges:

- Large measurement ranges;
- Sphere-like protrusions;
- Surface and sub-surface recesses and pores;
- Changing reflectivity;
- Large scales of interest; and
- Re-entrant features.

The standard continues on to provide guidance regarding surface preparation (i.e., by performing support removal and finishing operations), as well as the type of features that manufacturers are likely to find on such surfaces, whether they be top, side or bottom/supported surfaces.

The main bulk of the standard's guidance lies in its summary of the instruments commonly used for measurement of metal PBF surfaces. A summary of the principle of

operation and measurement good practice guidance is provided in regard to contact stylus measurements and optical measurements by imaging confocal microscopy, coherence scanning interferometry, focus variation microscopy and X-ray computed tomography. Methods for understanding measurement quality through examination of non-measurement points, repeatability error and measurement noise are also provided. Instrument calibration is also covered, in reference to ISO 25178-600 [37] and existing good practice guidance from the National Physical Laboratory [38–40].

Methods of measurement planning are the next primary topic covered by the standard, in reference to measurement location, repeats and sampling, as well as considerations regarding field of view sizes, field stitching, instrument resolution and instrument slope limitations. Bandwidth matching [41] is explained in detail, as a means of comparing the surface texture data acquired using different instruments by homogenising the band of spatial frequencies captured by multiple instruments and measurement setups.

Regarding characterisation good practice, the standard explains methods for determining filtering operations and choosing appropriate filtering values using either filter-stability or feature-dependent approaches. The relevant literature is simplified in the standard into understandable material and simple recommendations are given regarding good practice—for example, surface metrologists are recommended to acquire areas of at least $(2.5 \times 2.5)$ mm in size and to choose Gaussian convolution L-filters [42] between 250 μm and 800 μm for the characterisation of weld tracks. The standard suggests possible parameters that may be of use to those wishing to characterise metal PBF surfaces but emphasises the importance of individuals choosing filters to suit their specific requirements.

Extensive guidance is provided in the standard regarding the reporting of measurement results and data, focussing on the importance of reporting filter parameters alongside other aspects of the measurement and characterisation process.

## 3. Case Study

Here, we provide an example case study measurement of a metal PBF surface, following the good practice guidance presented in ASTM F3624-23 [36]. We present the case study in line with the sections presented in the standard, reporting various aspects of the measurement and characterisation pipeline in line with the sections of the standard, as a worked example of how the standard might be applied in a typical measurement.

### 3.1. Test Surface and Surface Preparation

An example metal PBF artefact was selected. In this case, a Ti-6Al-4V artefact manufactured using a Renishaw AM250 PBF-LB system using the manufacturer's nominal processing parameters for this material. The test surface was produced at an angle of 90° to the build direction, thus representing an example top surface, as discussed in ASTM F3624-23 [36]. After manufacture, the artefact was removed from the build plate and support structures were removed, both using pliers. No surface finishing operations were performed; the surface was left in the as-built state. The artefact was stored in a polythene bag between manufacture and measurement and was left to soak in the measurement laboratory environment (20 ± 0.5 °C, 50 ± 10% relative humidity) for 24 h prior to measurement. The test surface was cleaned using compressed air and acetone immediately prior to measurement. An image of the artefact is presented in Figure 1.

### 3.2. Instruments for Surface Texture Measurement of PBF Surface

The instrument was measured using two instruments, employing the objective lenses and measurement setups described in Table 1. Optimisation of both instruments was performed in accordance with guidance presented in ASTM F3624-23 [36] and in line with the findings presented in [43,44], for coherence scanning interferometry (CSI) and focus variation microscopy (FVM), respectively. Specific instrument names are omitted from this manuscript to prevent undue comparison of commercial instruments.

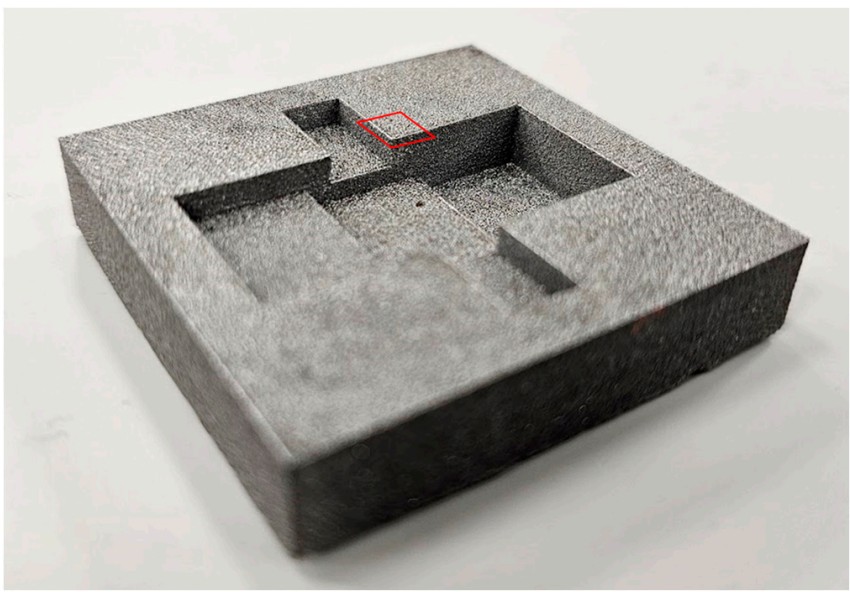

**Figure 1.** Image of the test surface used for this case study, with the approximate measurement area highlighted in red. The measurement area is an arbitrary field of approximately (2.5 × 2.5) mm.

**Table 1.** Measurement instruments and setups, where NA is numerical aperture, FoV is field of view and OR is optical resolution.

| Measurement Technology | Objective Lens Information | Measurement Setup |
|---|---|---|
| CSI | 5.5× lens, 1× zoom; NA: 0.15; FoV: (1.56 × 1.56) mm; and OR: 1.561 μm. | Acquisition time/measurement: 649 s; Vertical scan length: 300 μm (vertical stitching); Lateral stitching: 2 × 2 FoVs, auto-level performed before acquiring each frame; Topography reconstruction method: fringe analysis methods based on the coherence envelope; Source spectrum filtering: narrow-bandwidth; High dynamic range mode: two light levels; Signal oversampling: 4×; "Auto-tilt" and "auto-focus" performed at the beginning of the measurement; Laboratory temperature: 20.1 °C (start) 20.3 °C (end); Laboratory relative humidity: 52% (start) 52% (end); and Single measurement, five repeats taken immediately after one another. |
| FVM | 20× lens; NA: 0.4; FoV: (0.42 × 0.42) mm; and OR: 0.68 μm. | Acquisition time/measurement: 300 s (approx.); Vertical scan length: 412 μm; Vertical resolution: 100 nm; Lateral resolution: 2.94 μm; Illumination type: unpolarised coaxial light; Lateral stitching: 4 × 4 FoVs; Contrast: 0.50; "Auto-light" performed at the beginning of the measurement (brightness: 4.43 ms); Laboratory temperature: 20.3 °C (start) 19.6 °C (end); Laboratory relative humidity: 52% (start) 52% (end); and Single measurement, five repeats taken immediately after one another. |

### 3.3. Measurement Planning

Measurements were acquired in approximately the same location on the surface, using a corner of the test surface as a fiducial marker for approximate relocation of the surface with respect to both measurement instruments. For this example case study, a single location was selected for measurement, though five repeat measurements were taken at that measurement location. In industrial applications, it is common for a larger number of measurement locations to be selected, with a minimum of five locations considered to represent good practice. For both instruments, the five repeat measurements were taken immediately one after another, without repositioning the artefact between measurements. In line with the guidance, multiple fields of view were stitched (2 × 2, in both cases) to ensure sufficient resolution to capture weld ripple information, while covering a sufficiently large measurement area greater than (2.5 × 2.5) mm. Consideration was also given to the numerical aperture of the lenses employed in relation to slope limitations, in a trade-off against the size of the field of view and total measurement time. Measurements were optimised to minimise the number of non-measured points and outliers, so no outlier removal or non-measured point filling operations were deemed necessary.

### 3.4. Characterisation of Surface Texture

Once acquired, data were exported in their native file formats from the two measurement instruments and imported into Mountains 9 [45] surface texture characterisation software. The first CSI measurement was arbitrarily set as the reference and all CSI and FVM measurement were aligned to it in Mountains. An area of (2.5 × 2.5) mm, representative of the steady state of the surface (i.e., away from the corner of the test surface) was extracted from each dataset.

Bandwidth matching [41] was then performed. Firstly, the FVM measurements were decimated by a factor of three, from (0.44 × 0.44) μm to (1.31 × 1.31) μm, to approximately match the pixel spacing of the CSI measurements, at (1.56 × 1.56) μm. Then, the following operations were performed on all topography datasets to complete the bandwidth matching procedure:

- S-filter (high spatial frequency noise removal) with nesting index at 9 μm;
- Levelling by least-squares mean plane subtraction; and
- L-filter (waviness removal) with a nesting index at 0.8 mm.

The S-filter nesting index value was chosen to remove noise from both datasets, approximately equal to three times the lateral resolution of the lowest resolution dataset (in this case, the FVM data). The value of the lateral resolution used here was 2.94 μm, relating to the value used by the local-contrast-finding algorithm employed by the FVM instrument's software (see Table 1). The L-filter was chosen to remove underlying waviness effects larger than the width of two weld tracks. The ISO 25178-2 [46] *Sq* surface texture parameter (the root mean square height of the scale-limited surface) was finally generated for all measured datasets.

### 3.5. Reporting of Measurement Results and Data

In Figure 2, the results of this brief case study are presented, showing similar, but statistically different, values acquired for the *Sq* surface texture parameter using two different instruments. These figures are not in statistical agreement, most likely due to effects explored extensively elsewhere [2], though further study of the specific surface is required to determine exactly why the parameter results are significantly different.

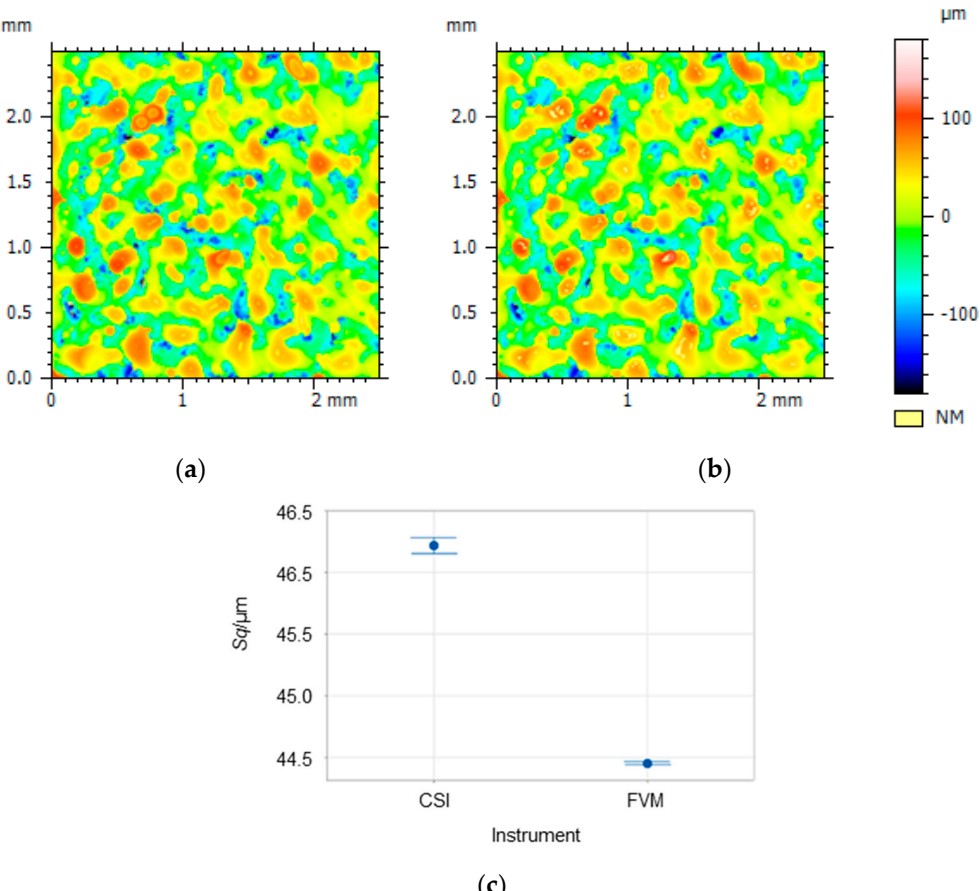

**Figure 2.** (**a**) Example filtered surface dataset (CSI); (**b**) example filtered surface dataset (CSI); and (**c**) the *Sq* parameter, as acquired using CSI and FVM, presented at 95% confidence, calculated from five repeat measurements using t-distributions [47]. "NM" are non-measured points.

## 4. Discussion and Future Additions to the Standard

The aims of this paper are twofold—firstly, to introduce ASTM F3624-23 [36] to the academic community and demonstrate its use via a simple case study; and, secondly, to explain the likely next steps for the research community within the context of the industrial standard. While the standard provides guidance for some of the most common surface texture measurement instruments, there are an increasing number of alternative and novel measurement technologies not yet covered in the current edition of the standard. For example, while extensive research has been performed in CSI and FVM measurement of metal PBF surfaces, there has not been similar effort made in developing good practice for measurements made using chromatic confocal methods [48], point autofocus instruments [49] or more novel surface measurement systems such as GelSight [50]. Each technology requires further investigation, as will technologies which have not yet been invented.

It is also important to note comments raised by the ASTM community through the publication of the first edition of the standard. In this first edition, the focus was placed on providing good practice for the most common means of characterising surfaces, particularly via ISO 21920-2 [31] and 25178-2 [46] parameters. However, other methods of characterisation not covered in this first edition are growing in popularity, most notably feature-based methods (for example, see [1,51,52]), but also multi-scale methods [53] and many other possible analysis paradigms. Some early examples of good practice for these characterisation paradigms exist in the literature (e.g., see [1,52]), but standardised good practice remains somewhat further down the line in these cases. These issues all represent possible future additions to ASTM F3624-23 [36] and the community should remain cog-

nisant of ongoing developments to ensure that such developments are included in future iterations of the standard.

**Author Contributions:** Conceptualization, R.L.; methodology, A.T. and L.N.; formal analysis, A.T. and L.N.; investigation, A.T. and L.N.; resources, A.T. and L.N.; data curation, A.T. and L.N.; writing—original draft preparation, A.T.; writing—review and editing, L.N. and R.L.; visualization, A.T. and L.N.; supervision, R.L.; project administration, R.L.; funding acquisition, R.L. All authors have read and agreed to the published version of the manuscript.

**Funding:** This research was funded by the UKRI Research England Development (RED) Fund via the Midlands Centre for Data-Drive Metrology. This research was also funded by the Engineering and Physical Sciences Research Council (Grants EP/L01534X/1 and EP/M008983/1) and the Manufacturing Technology Centre (Coventry, UK).

**Data Availability Statement:** Data are available in a publicly accessible repository. The data presented in this study are openly available in Zenodo at [54].

**Acknowledgments:** The authors would like to acknowledge the publications written, conference presentations delivered and documentary review work by members of the global surface topography measurement community, as well as valuable conversations with. The community was extremely valuable in the creation of this standard and we would like to thank them all for their engagement and support throughout its development.

**Conflicts of Interest:** The authors declare no conflict of interest.

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
