# Peer review of "New Standard for Metal Powder Bed Fusion Surface Texture Measurement and Characterisation"

_2673-8244, doi:10.3390/metrology3020013_

Round 1

Reviewer 1 Report

1. Please add the main PBF parameters.

2. Why is the red area in Figure 1 selected for the case study?

3. Please compare the advantages and disadvantages between CSI or FVM.

Author Response

Please see the attached response document.

Reviewer 2 Report

The authors show the partial practical application of a recently adopted ASTM standard related to the characterization of metal surfaces built by additive manufacturing. The scientific-theoretical contribution of the work is minimal. However, the technical-experimental contribution is outstanding since it puts into practice for the first time the implementation of a new standard that is of outstanding importance for the new metal-mechanic techniques of additive manufacturing.

The manuscript is well written and organized. I only have the following minor comments that must be attended to before possible acceptance:

1. The artefact presented in figure 1 lacks labels that allow it to be sized. One might think that the red box is roughly 2.5 x 2.5mm, but it's not clear that this is the case. I recommend putting dimensions or a scale bar.

2. Homogenize the font, type and size, of the axes and labels in figure 2.

3. From the same figure 2, I cannot identify the meaning of the NM label and its respective pale-yellow rectangle. If it is important, indicate it in the figure caption, if it is not, it is better to remove it.

4. About reference 21. The indicated year of publication is 2019, but both the number including the name (F3624-23) and the standard itself indicate that it was adopted or revised in 2023. Please clarify this information in the manuscript.

Best Regards.

Author Response

(The authors gave the same response as above.)

Reviewer 3 Report

This manuscript provided a comprehensive introduction of a new standard guide, ASTM F3624-23, for measuring and characterizing surface texture in AMed metallic materials via PBF. The authors highlight the challenges in accurately measuring and characterizing complex surface features in metal AM and emphasize the need for such a standard. A case study is also presented to demonstrate the application of the standard guide: measurement of a metal PBF surface using two different instruments following the good practice guidance. The paper also discusses future considerations for the standard, including the need to incorporate novel measurement technologies and feature-based characterization approaches.

My only suggestion to the authors is to discuss more the benchmarking of the new standard. Readers of this paper would likely be interested in understanding its advantages compared to other measurement methods in this field.

In general, this manuscript is well-written and all contents are clearly presented. I recommend to publish this manuscript in Metrology

Author Response

(The authors gave the same response as above.)
